# MODELPIRATE: SECURITY ANALYSIS OF PARTIAL MERGING AGAINST MODEL STEALING ATTACKS

## ABSTRACT

Model merging is a promising technique to enhance the capabilities of neural networks (NNs) by integrating multiple downstream fine-tuned models without requiring access to clients' raw data or substantial computation resources. However, conventional model merging typically requires collecting the full set of fine-tuned parameters from multiple clients, which may expose them to model-privacy risks. An emerging approach, known as *partial model merging (PMM)*, mitigates this risk by splitting the model into private and shared parts, where only the shared part is merged while the private part remains local to each client. Despite its stricter parameter fusion, PMM can still achieve competitive performance compared to full-parameter sharing. However, the privacy properties of PMM remain underexplored. In this paper, we propose a novel model stealing attack and assess the risk of reconstructing the unshared private part of a partially merged model under eight attack scenarios with varying prior knowledge (i.e., partial training data, model parameters and/or model structure). Our comprehensive experiments reveal that merging NNs without adequate protection is highly vulnerable. Even when only a small fraction of the training data, model parameters, or model structure is exposed, adversaries can still recover significant portions of the private model's performance.

## 1 INTRODUCTION

*Model merging* (aka model fusion) (Yang et al., 2024b; Yadav et al., 2023; Xu et al., 2024) integrates multiple downstream fine-tuned neural network (NN) models with diverse capabilities into a single model without retraining or additional fine-tuning. It enables effective reuse, fusion, and transfer of users' knowledge. Hence, users without relevant domain-specific data can mutually benefit from other users who have the data without exchanging their raw data. A widely adopted approach of multi-task model merging is the Task Arithmetic method introduced by Ilharco et al. (2023), where multiple vectorised models (i.e., task vectors) are summed to produce a single merged model. This group of approaches requires collecting the complete set of fine-tuned parameters from multiple entities and then merging these parameters to construct a universal merged model. It is known as full model merging (FMM), as depicted in Fig. 1a.

However, the domain-specific fine-tuned models are increasingly proprietary and closed-source due to the high costs of data collection and training, making the distribution of full parameter sets impractical in many real-world FMM deployments. Moreover, FMM compromises **model privacy**. An adversary can perform *Model stealing attacks* by constructing an alternative NN model (i.e., a surrogate model) that closely mimics behaviours of the victim model (Papernot et al., 2017; Orekondy et al., 2019; Roberts et al., 2019), thereby obtaining a local copy that substitutes for the original victim model without incurring additional cost.

In response, *partial model merging (PMM)* (Stoica et al., 2024) has emerged as a viable alternative, wherein the full model is partitioned into private and shared parts, as illustrated in Fig. 1b. PMM

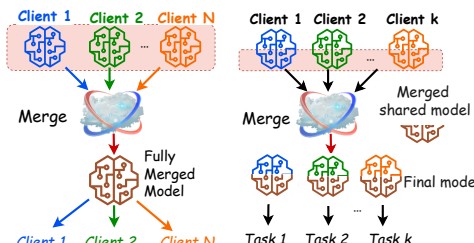

(a) Full model merging (b) Partial model merging

Figure 1: Full model merging versus Partial model merging. (a) A merged model for $N$ tasks. (b) Assemble $N$ partitioned models with a merged model part (shared part) for $N$ tasks.

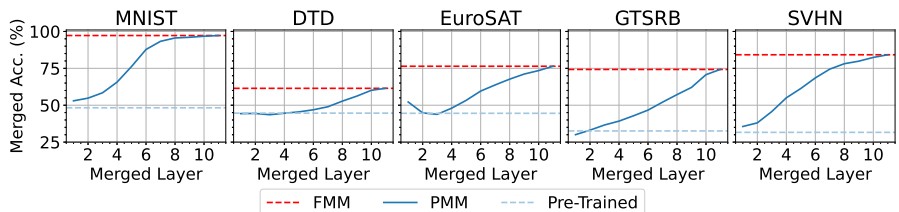

Figure 2: Merged ViT-B/32 model accuracy for the partially merged model with different numbers of merged layers. The merging is performed between models fine-tuned on five different downstream tasks (i.e., MNIST, DTD, EuroSAT, GTSRB, and SVHN).

enforces a stricter parameter fusion and only merges the shared parts of the model, while the private parts remain with local clients. This design reduces the number of model parameters shared and therefore reduces both overheads and the potential for model-privacy leakage.

Empirically, we observed that PMM can achieve higher model performance than the pre-trained model and closer to FMM when a larger portion of the model is merged. To illustrate this phenomenon, we use the widely adopted ViT-B/32 model (Radford et al., 2021) and evaluate across five benchmark datasets (i.e., MNIST, DTD, EuroSAT, GTSRB, and SVHN). Specifically, we analyse the accuracy of both partially and fully merged models across multiple tasks by varying the number of merged transformer layers. As shown in Fig. 2, the red dashed lines indicate the FMM performance, serving as empirical upper bounds, while the light blue dashed lines indicate the pre-trained model performance before fine-tuning, serving as lower bounds. The solid blue curves trace PMM accuracy as the number of merged transformer layers increases. We observe that the accuracy of the merged model generally increases as more layers are merged across all five datasets. For ViT-B/32, merging 75% of layers from downstream fine-tuned models yields PMM that retains at least 85.89% of the accuracy of FMM, while reducing communication and computation costs to about 75% of FMM's costs. Similar trends can be observed for ViT-B/16 and ViT-L/14, with detailed results provided in Appendix A.

Although PMM limits model exposure by sharing only a subset of layers, the potential model privacy risks associated with this approach remain unexplored. In particular, it is unclear to what extent sharing parameters incurs model privacy leakage. To the best of our knowledge, no existing work has examined potential model privacy vulnerabilities under PMM, nor the privacy-utility trade-off induced by varying the number of shared parameters. These gaps motivate our central question:

**How would model privacy be affected by sharing a subset of model parameters for merging?**

To make this question concrete, we quantify model privacy risk in terms of how successfully an adversary can extract private model behaviour under PMM. We identify a two-sided information asymmetry: On the adversary side, the victim's training samples, model structure and parameters are largely hidden, which constrains attack design and makes evaluation difficult under realistic assumptions; On the victim side, the adversary's objectives and capabilities are often unknown, which prevents direct measurement of leakage risk and decide the number of layers shared in PMM to balance generalisation and privacy exposure. With this framing, we assess model-privacy risks from the adversary's perspective under different knowledge constraints. The detailed contributions of this paper are as follows:

- We perform the first-of-its-kind systematic privacy analysis of PMM.

- We introduce *ModelPirate*, a model-stealing attack tailored to PMM. The proposed *ModelPirate* aims to recover the behaviour of the private part of the model given limited prior knowledge.

- We evaluate *ModelPirate* in eight attack scenarios with prior knowledge across diverse models and datasets. Our results offer empirical guidance for attack defence and client layer-sharing decisions.

## 2 PRELIMINARIES

In this section, we formally define PMM. It is commonly known that in an NN model, the layers closer to the outputs contain information that is more specific to the model's tasks (Nasr et al., 2019; Vandenhende et al., 2022). Therefore, we consider PMM clients sharing layers closer to the inputs,

while keeping the rest of the layers closer to the outputs private to improve the merged model's generalisation while keeping the task-specific information private. Let a pre-trained base model be $f_\theta : \mathcal{X} \to \mathcal{Y}$ with $L$ layers and parameters $\theta = (\theta^1, \theta^2, \ldots, \theta^L)$. Each client separates its full individual model into two parts at layer $l$: the *private part* $P(\theta) := \theta^{l+1:L}$ remains private at the client, whereas the *shared part* $S(\theta) := \theta^{1:l}$ is sent to the a merging entity or uploaded to a platform such as Hugging Face [1] for partial open-source. We assume all clients share the same $l$ for merging feasibility.

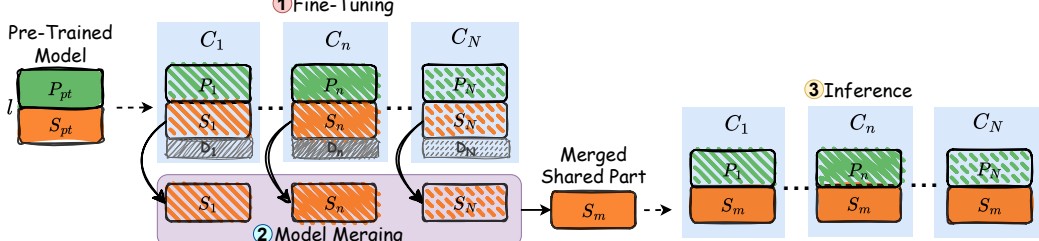

Figure 3: Partial model merging. Client $C_n$ fine-tunes the pre-trained model and partitions its fine-tuned model into private part $P_n$ and shared part $S_n$. The merged shared part $S_m$ is connected to the private part $P_n$ at client $C_n$ for inference. Dashed arrows indicate model distributions where the pre-trained/merged model is distributed to all clients $C = \{C_n | n = 1, \cdots, N\}$.

Let the total number of participating clients be $N$. As illustrated in Fig. 3, each client $C_n \in C$ fine-tunes a common pre-trained model $S_{pt} + P_{pt}$ on their own task $T_n$ with data $D_n$, obtaining $\theta_n$ and thus shared part $S_n := S(\theta_n)$, private part $P_n := P(\theta_n)$ (Step 1). Then, the shared parts from all clients $\{S_n\}_{n=1}^N$ are merged as $S_m = \mathcal{M}(S_1, \ldots, S_N)$, the *merged shared part* [2], using the merging algorithm $\mathcal{M}$ (Step 2). Finally, client $C_n$ uses the obtained partially-merged model $S_m + P_n$ for inference (Step 3).

## 3 OUR ATTACK: *ModelPirate*

### 3.1 PROBLEM DEFINITION

We study a benchmark adversarial setting under PMM with a single adversary-victim pair. As illustrated in Fig. 4, client $C_a$ is the *adversary* and client $C_v$ is the *victim*, where $a \neq v$. The two clients are fine-tuned on different downstream tasks, i.e., $T_a \neq T_v$. The adversary is *honest but curious*. It follows the PMM protocol as an ordinary participant while attempting to reconstruct the victim's private model. Following the notation in Sec. 2, victim $C_v$ holds a fine-tuned model split at layer $l$ into $(S_v, P_v)$. We define the *target model* as

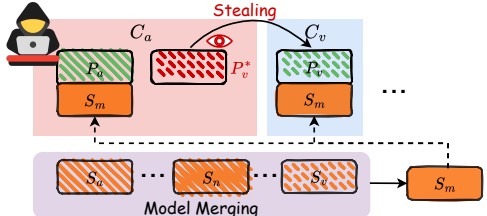

Figure 4: Adversarial model. The adversary $C_a$ trains a clone model $P_v^*$ to simulate the behaviour of the victim's $C_v$ private part $P_v$.

$$f_v(x) := f(x; S_v, P_v). \tag{1}$$

Specifically, we adopt a partially homogeneous PMM setting where all shared parts to be merged $\{S_n\}_{n=1}^N$ are structurally compatible, whereas private parts $\{P_n\}_{n=1}^N$ may be heterogeneous. The adversary's goal is to construct a *clone private model part* $P_v^*$ such that the composed model

$$\tilde{f}_v^\star(x) := f(x; S_v, P_v^*) \tag{2}$$

---

[1] https://huggingface.co

[2] In this paper, we consider a general PMM scenario, where the $S_n$ from each client $n$ is sent to a merging entity. The merging entity can be one of the participating clients or a third party (e.g, a cloud server). Then, model merging will be performed at the merging entity after it receives $S_n$ from all clients.

mimics the behaviour of $f_v(x)$. In particular, for a query distribution $Q$ (e.g., induced by accessible data), we aim for a behavioural discrepancy $\mathbb{E}_{x \sim Q}\left[d\left(f_v(x), \tilde{f}_v^\star(x)\right)\right]$ to be minimal, where $d(\cdot, \cdot)$ denotes a task-appropriate distance.

### 3.2 Threat Model

#### 3.2.1 Adversary's prior knowledge

We distinguish three types of prior knowledge (i.e., *Self-knowledge* $\mathcal{K}_{\text{self}}$, *Shared knowledge* $\mathcal{K}_{\text{shared}}$ and *Auxiliary knowledge* $\mathcal{K}_{\text{aux}}$) available to $C_a$:

- $\mathcal{K}_{\text{self}}$: information inherently possessed by $C_a$, including it's full *source model* $S_a + P_a$, its training data $D_a$ for task $T_a$, and access to the pre-trained model $S_{pt} + P_{pt}$ before fine-tunning.
- $\mathcal{K}_{\text{shared}}$: artefacts made visible by the PMM protocol. In particular, the merged shared part, $S_m = \mathcal{M}(S_1, \ldots, S_N)$, is available to participating clients as a white box. Additionally, $C_a$ can query the victim's full model, $S_v + P_v$, as a black box without knowing its model structure and parameters. [3] [4].
- $\mathcal{K}_{\text{aux}}$: optional side information beyond the above. We regard it as being structured along three auxiliary axes, each of which directly determines the composed clone function: $\tilde{f}_v^\star(x) = f(x; S_v, P_v^*)$. (i) white-box access to $S_v$ (e.g., available if $C_a$ is a merging entity, or released via partial open-source); (ii) a structural prior $M_{P_v}$ about $P_v$ where the model parameters are unknown, and (iii) a subset $\hat{D}_v \subset D_v$ of victim data, where $|\hat{D}_v| = p_d \times |D_v|$ and $p_d$ is the proportion of $C_v$'s training data available to $C_a$.

The first two categories, $\mathcal{K}_{\text{self}}$ and $\mathcal{K}_{\text{shared}}$, are *protocol-compliant* and typically available to any honest PMM participant. Our analysis therefore centres on $\mathcal{K}_{\text{aux}}$, systematically varying the availability of $(S_v, M_{P_v}, \hat{D}_v)$ because these three components directly parameterise the clone $\tilde{f}_v$. For clarity, we encode the presence of these auxiliary axes via indicators:

$$\mathbb{I}_s = \left\{ \begin{array}{ll} 0 & S_v \text{ is unknown} \\ 1 & S_v \text{ is known} \end{array} \right. , \mathbb{I}_p = \left\{ \begin{array}{ll} 0 & M_{P_v} \text{ is unknown} \\ 1 & M_{P_v} \text{ is known} \end{array} \right. , \mathbb{I}_d = \left\{ \begin{array}{ll} 0 & D_v \text{ is unknown} \\ 1 & D_v \text{ is known} \end{array} \right. \tag{3}$$

We identify eight attack scenarios based on the varying levels of $\mathcal{K}_{\text{aux}}$ available to the adversary to examine how different degrees of information exposure influence the feasibility and effectiveness of potential attacks. For simplicity, we denote the attack scenarios as $\mathcal{AS}[\mathbb{I}_s \cdot \mathbb{I}_p \cdot \mathbb{I}_d]$ for the rest of this paper. For example, $\mathcal{AS}[000]$ means that none of $S_v$, $M_{P_v}$, or $D_v$ is known to the adversary.

#### 3.2.2 Adversary's objective

We define the model accuracy as the performance achieved on task $T_v$ using the validation set.

- **Local accuracy** is the model accuracy of $C_v$'s local pre-merged fine-tuned $S_v + P_v$ (i.e., target model). It serves as the **upper bound accuracy** of the clone model as it reflects the performance of the model fine-tuned solely on $T_v$.
- **Merged accuracy** is the model accuracy of the partially-merged model $S_m + P_a$. This accuracy serves as the **lower bound accuracy** of the clone model. Note that the merged model is a multi-task model, which is expected to yield lower performance than the fine-tuned single-task models on $T_v$ due to interference between different tasks.
- **Clone accuracy** is the model accuracy of the full clone model $S_v + P_v^*$. It is the **realised accuracy** that directly measures the effectiveness of the model stealing attack. The clone model is a single-task model dedicated to $T_v$. Therefore, the clone accuracy is expected to be higher than the merged accuracy.

---

[3] A practical example of this black-box attack is that the adversary queries a **commercially available model** and uses the responses to reconstruct the proprietary model parameters (Krishna et al., 2020). In this case, after a limited number of queries and a model extraction process, the adversary can maintain their own copy of the model and use it without incurring any further costs to the original model owner.

[4] Note that, different from the conventional black-box attacks, the adversary in the *ModelPirate* attack has additional prior knowledge and can therefore construct a clone model with a better performance depending on the PMM setup, which we will discuss in the following sections of this paper.

$C_a$'s objective is to construct a clone private part $P_v^*$ such that the clone accuracy is as high as possible. This would improve the model accuracy that $C_a$ can achieve on $T_v$ using the full clone model $S_v + P_v^*$ or $S_m + P_v^*$, compared to the cases where $C_a$ performs task $T_v$ using $S_v + P_a$ or $S_m + P_a$. [5]

### 3.3 TRAINING THE CLONE MODEL

We present the overall training procedure of the clone model in Fig. 5. The adversary $C_a$ composes a clone model by *freezing* a shared part $k_s$ and optimising only the private part $P_v^*$ on data $k_d$, thus:

$$\tilde{f}_v^{\star}(k_d) := f(k_d; k_s, P_v^*),$$
$$k_d \in \{\hat{D}_v, \hat{D}_a\}, \quad k_s \in \{S_v, S_m\}. \quad (4)$$

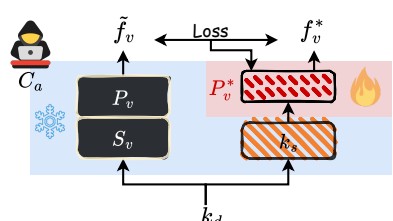

Figure 5: Clone model training. $P_v + S_v$ is a black box, $k_s \in \{S_v, S_m\}$ is a frozen white box, and only the cloned private part $P_v^*$ is trained during the attack using $k_d \in \{\hat{D}_v, \hat{D}_a\}$.

**For shared part $k_s$ of clone model.** We treat $k_s$ as a *frozen* module, i.e., $\nabla_{\theta^{1:l}} f(k_d; k_s, \cdot) = \mathbf{0}$. The choice of $k_s$ depends on whether the victim's shared part is available ($\mathbb{I}_s \in \{0, 1\}$). Specifically,

- $\mathbb{I}_s = 0$: $C_a$ has no direct knowledge of the victim's shared model part. In this case, the merged shared model $S_m$ is the only model part that embeds task-specific knowledge for $T_v$, and thus we set $k_s = S_m$.
- $\mathbb{I}_s = 1$: under fully distributed merging, $C_a$ can obtain the victim's shared model part $S_v$. Since $S_v$ encodes $T_v$ without task-interference from other clients, it is prioritised over $S_m$, and thus we set $k_s = S_v$.

**For private part $P_v^*$ of clone model.** $P_v^*$ is regarded as the *trainable* module, i.e., $\nabla_{\theta^{1:l}} f(k_d; k_s, P_v^*) \neq \mathbf{0}$. The structure of $P_v^*$ depends on whether the victim's private-structure is known ($\mathbb{I}_p \in \{0, 1\}$):

- $\mathbb{I}_p = 1$: $C_a$ knows $M_{P_v}$ and $P_v$ is architecturally homogeneous with $P_a$ and $P_{pt}$. We initialise $P_v^*$ from the pre-trained parameters $P_{pt}$ to accelerate convergence and better preserve the inductive bias of $P_v$.
- $\mathbb{I}_p = 0$: $C_a$ lacks knowledge of $M_{P_v}$. To mitigate overfitting under limited training data while retaining sufficient expressivity, we adopt a *deep–shallow* design described below.

For $\mathbb{I}_p = 0$, we assume that $P_v$'s model structure is relatively complex, and a model with a structure similar to $P_v$ can be trained to capture the behaviour of $P_v$. Therefore, we construct a deep sub-model $M_1$ to ensure that the clone model has sufficient complexity to simulate the behaviour of $P_v$. In parallel, a shallow sub-model $M_2$ bypasses $M_1$ to avoid overfitting and improve the clone model's generalisability, as some $C_v$'s private training data is unseen by $C_a$. As shown in Fig. 6, the inputs of $P_v^*$ are also the inputs of both $M_1$ and $M_2$, and a dense layer $M_3$ connects the concatenated $M_1$ and $M_2$ outputs to the outputs of $P_v^*$. Depending on $C_a$'s prior knowledge of $C_v$'s model, the deep sub-model $M_1$ can leverage any deep model structure that behaves similarly to $P_v$ (e.g., LSTMs to

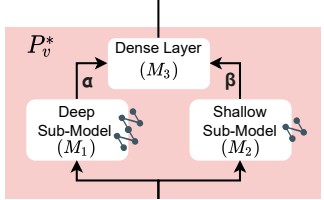

Figure 6: Internal structure of $P_v^*$ for $\mathbb{I}_p = 0$. $\alpha$ and $\beta$ are the weight of models $M_1$ and $M_2$, respectively.

simulate transformers) with a similar number of layers and neurons. For the worst-case scenario that $C_a$ has zero knowledge about $C_v$'s model structure, $M_1$ should follow a similar structure to $P_a$.

To evaluate the cloned model's task-specific performance, we connect the classification head for task $T_v$ to the output of $P_v^*$. Note that the classification head cannot be merged, and it remains unchanged during the model stealing process.

**For training data $k_d$ on clone model.** Inputs $k_d$ are fed to both $k_s + P_v^*$ and the black-box of $S_v + P_v$. We denote their outputs as $\tilde{f}_v^*$ and $f_v$, respectively. The choice of $k_d$ is determined by the data-availability indicator $\mathbb{I}_d \in \{0, 1\}$:

---

[5]Note that the validation set and the training set have no overlapping data samples. The validation set is unknown to $C_a$, and the clone model $P_v^*$ is unknown to $C_v$. Therefore, we measure the clone accuracy to evaluate the attack success rate, which cannot be assessed by either $C_a$ or $C_v$.

$$k_d = \begin{cases} D_a, & \text{if } \mathbb{I}_d = 0 \quad \text{(no victim samples available)}, \\ \hat{D}_v, & \text{if } \mathbb{I}_d = 1 \quad \text{(use victim subset } \hat{D}_v \subset D_v \text{ with } |\hat{D}_v| = p_d \times |D_v|). \end{cases} \quad (5)$$

When $\mathbb{I}_d = 1$, $\hat{D}_v$ is prioritised as it is directly aligned with $T_v$.

**Optimisation of** $P_v^*$. The adversary's objective is to align behaviours of $k_s + P_v^*$ and $S_v + P_v$ on the same inputs. We optimise only $P_v^*$, keeping $k_s$ frozen, by minimising a pointwise discrepancy between outputs. We adopt MAE in this paper for its simplicity and effectiveness, while other metrics, such as CE and KL, can also be applied.

$$\mathcal{L}_{atk} = \frac{\sum\limits_{i \in k_d} \left| \tilde{f}_v^*(i; k_s, P_v^*) - f_v(i, S_v, P_v) \right|}{\left| \tilde{f}_v^*(i; k_s, P_v^*) \right|}. \quad (6)$$

## 4 EXPERIMENTS

### 4.1 EXPERIMENTAL SETUP

Unless otherwise specified, we use the default hyperparameters listed in Appendix B for the experiments. For consistency, we consider the layer-wise Task Arithmetic (Ilharco et al., 2023) as the default PMM algorithm in this paper and use the corresponding datasets and models for evaluation. Table 2 in Appendix A shows that when 75% of the layers are merged, the difference between FMM and PMM is less than 10%. Therefore, we set the default proportion of the PMM merged layer to be 75%. In addition to the default image classification datasets and vision transformer models used in model merging (Ilharco et al., 2023), we extend our experiments to Natural Language Processing (NLP) tasks using the IMDB (Maas et al., 2011) and QASC (Khot et al., 2019) datasets with T5 model to show the generalisation of the *ModelPirate* attack beyond the previously considered computer vision models and datasets. The setup for the extended experiments will be detailed in Appendix C.

**Dataset.** Table 3 in Appendix D lists the datasets we consider for model merging. While the models fine-tuned for different datasets are used for merging, we will focus on the DTD and EuroSAT datasets as their input features have similar properties (i.e., patterns of different textures and landscapes) while the classification tasks and difficulties differ. We repeat the attack simulations with MNIST and SVHN datasets and present the results in Appendix E. We note that there are different numbers of data samples available for each dataset. Therefore, for a fair comparison, we ensure that the number of data samples per class (i.e., see Avg. values in Table 3, Appendix D) is similar across all datasets by randomly selecting a subset of data samples in the "Original" dataset as the "Adjusted" dataset.

**Model.** We consider the Contrastive Language-Image Pre-training (CLIP) (Radford et al., 2021) model with a Vision Transformer (ViT) as the image encoder and a Transformer-based text encoder, following the same model structures in previous model merging literature (Radford et al., 2021; Ilharco et al., 2023), namely ViT-B/32, ViT-B/16 and ViT-L/14. The three pre-trained models are fine-tuned on the five datasets listed in Table 3, using the default setups in (Ilharco et al., 2023).

### 4.2 EXPERIMENTAL RESULTS

We present the results evaluated on the DTD and EuroSAT datasets. Unless otherwise specified, we denote the clone models as $D_a \rightarrow D_v$, where $D_a$ and $D_v$ are the datasets for fine-tuning the source and target models, respectively. To reduce the impact of outliers while ensuring reproducibility, we repeat the experiments with five different random seeds and present the average values as the results.

### 4.3 OVERALL PERFORMANCE EVALUATION OF *ModelPirate*

For benchmark comparison, we consider existing state-of-the-art query-based model stealing attacks that match our attack scenarios, namely Knockoff (Orekondy et al., 2019), JBDA (Papernot et al., 2017) and Random (Roberts et al., 2019). The three existing model stealing attacks were designed for attack scenarios similar to that described in $\mathcal{AS}$[100], $\mathcal{AS}$[101] and $\mathcal{AS}$[110], respectively. Table 1 lists the accuracies of clone models derived using *ModelPirate* under different attack scenarios

Table 1: Model accuracies of the clone models generated by our proposed *ModelPirate* attacks with the default hyper-parameters with benchmark comparison with Knockoff, JBDA and Random model stealing techniques. Note that the target model ViT-L/14 is more complex than ViT-B/16, which is more complex than ViT-B/32. Bolded numbers indicate success model stealing attacks where the clone accuracy surpasses the merged accuracy.

| Attack Method | EuroSAT→DTD | | | DTD→EuroSAT | | |
|---|---|---|---|---|---|---|
| | ViT-B/32 | ViT-B/16 | ViT-L/14 | ViT-B/32 | ViT-B/16 | ViT-L/14 |
| **Merged Acc.** | 52.77% | 54.84% | 72.50% | 67.67% | 79.11% | 95.04% |
| Knockoff | 7.18% | 7.18% | 2.13% | 52.22% | 57.22% | 16.70% |
| JBDA | 3.88% | 3.56% | 3.19% | 23.48% | 29.33% | 18.63% |
| Random | 2.93% | 2.13% | 35.37% | 10.44% | 12.67% | 38.70% |
| $\mathcal{AS}$[000] | 2.39% | 2.45% | 1.91% | 15.74% | 17.89% | 14.74% |
| $\mathcal{AS}$[100] | 37.27% | 8.54% | 2.44% | 53.74% | 31.07% | 23.61% |
| $\mathcal{AS}$[010] | 2.66% | 2.45% | 65.37% | 48.52% | 54.70% | **93.81%** |
| $\mathcal{AS}$[001] | 18.88% | 26.22% | 20.53% | 60.07% | 60.04% | 59.67% |
| $\mathcal{AS}$[101] | **68.35%** | **60.79%** | 31.23% | **96.83%** | **95.81%** | 79.37% |
| $\mathcal{AS}$[011] | 49.89% | **56.81%** | 82.82% | 65.52% | 68.22% | 73.37% |
| $\mathcal{AS}$[110] | **85.15%** | **78.40%** | **97.87%** | **98.38%** | **98.93%** | 52.54% |
| $\mathcal{AS}$[111] | **62.89%** | **65.42%** | **98.19%** | **96.34%** | **96.52%** | 63.22% |

$\mathcal{AS}$[XXX] and target model structures. Note that the default setting requires only about 100 queries to perform attacks.

From the results, we see that in most of the cases where there are at least two of $\mathbb{I}_s$, $\mathbb{I}_p$, or $\mathbb{I}_d$ present, the clone accuracy for *ModelPirate* surpasses or is close to the merged model accuracy. Generally, a simpler target task (i.e., EuroSAT) yields higher clone accuracy, whereas a more complex target task with more classification classes (i.e., DTD) yields lower clone accuracy. From these observations, we conclude that our proposed *ModelPirate* attack substantially outperforms the existing baselines under the same attack scenarios, target model structures and tasks. Generally, *ModelPirate* attack is **more effective for a less complex target task**.

Interestingly, for ViT-B/32 and ViT-L/14 models, $\mathcal{AS}$[110], where the victim's exact model structure is unknown to the adversary, outperforms $\mathcal{AS}$[111], with the adversary having the same target model structure as the victim and full prior knowledge on the victim's shared model parameters and partial training data. This shows the **advantage of $P_v^*$ we constructed** in Fig. 6 compared to the original model structure of the private part when the target model structure is relatively simple. We further explore this observation in the following experiments.

The increase in clone accuracy from $\mathcal{AS}$[X01] to $\mathcal{AS}$[X10] shows that knowing part of the victim's training samples would help the adversary to gain more in its attack performance than knowing the exact model structure of the victim's private part. Therefore, under the default settings, it is **more important for a client to protect its training data than its private model structure**.

The significant increase in clone accuracy from $\mathcal{AS}$[01X] to $\mathcal{AS}$[10X] shows that knowing the victim's shared model part would help the adversary to gain more in its attack performance than knowing the victim's private model structure. Therefore, under the default settings, it is **more important for a client to protect its shared model part than its private model structure**. It is suggested that a client should send its shared model part to a trusted merging entity to protect its model privacy.

We also observe that for a more complex model, the clone accuracy increases from $\mathcal{AS}$[1X0] to $\mathcal{AS}$[0X1]. The results show that knowing the victim's shared model part would help the adversary gain more in its attack performance than knowing the victim's training data, because of a large volume of information embedded in the complex model. Therefore, for a more complex model, it is **more important for a client to protect its shared model part than its training data**. For these cases, we suggest that a client should send its shared model part to a trusted merging entity to protect its model privacy. On the other hand, if the victim's model is simpler, an adversary can clone a model with better performance using a subset of the victim's dataset, even without any knowledge of the victim's

public model. Therefore, **protecting the training data is more important for a client to reduce model privacy leakage**.

### 4.4 Privacy-utility tradeoff: impact of the number of merged layers

Fig. 2 in Sec. 1 shows that in a partially-merged setup, the merged model accuracy for each individual task increases as the number of layers merged increases (i.e., a larger $l$). However, this comes at the cost of reducing privacy in terms of the difficulties of reconstructing the behaviour of the model part that is intended to remain private, as a larger volume of information can be obtained by the adversary. In this experiment, we change the separation layer $l$ and set the rest of the parameters as defaults to demonstrate this hypothesis qualitatively and quantitatively. We repeat the experiment for $\mathcal{AS}[101]$ and $\mathcal{AS}[111]$ and present the results as "Known Model" and "Unknown Model" correspondingly.

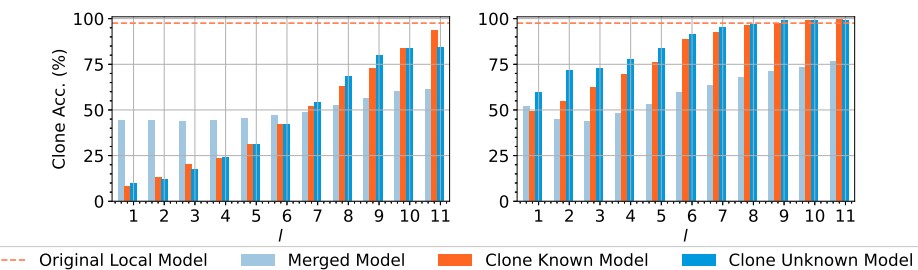

Figure 7: Clone accuracy for different numbers of merged layers $l$ in EuroSAT→DTD (left) and DTD→EuroSAT (right) scenarios.

The experimental results in Fig. 7 show that the general trends of clone model accuracies under the assumptions of known or unknown target model structure increase as the number of merged layers (i.e., a larger $l$) increases. We also see that the increase in the clone model accuracy is more significant in the EuroSAT→DTD scenario, where the target model performs a more difficult task than the source model.

As shown in Fig. 7 (left), the clone accuracy is less than the merged model accuracy when $l < 7$. The target model's clone accuracy surpasses the merged model accuracy at $l = 7$. Similarly, Fig. 7 (right) shows the DTD→EuroSAT scenario where the source model performs a more difficult task than the target model. The clone accuracy under this scenario is always higher than the merged model accuracy at the same separation layer, with the exception of layer one, where the known model clone accuracy is slightly lower than that of the merged model accuracy. Interestingly, we see that the clone accuracies for the last few layers (i.e., $l > 7$) are close to or even greater than the original local model's accuracy. This shows that merging more than seven layers would create a significant privacy vulnerability in the private part's model behaviour. We also conclude that the model stealing attack would be more successful if cloning a target model for a less difficult task.

### 4.5 Impact of the proportions of known data samples

Next, we focus on how the proportion of data samples the adversary can obtain from the victim affects the clone accuracy. We use subsets of the data samples from the "Adjusted" dataset (see Table 3) to ensure that the total number of data samples from each class is similar.

Fig. 8 shows that from when $p_d$ is between 20% and 100%, the increase in $p_d$ results in an increase in the clone model accuracy for both known and unknown models. The clone model accuracy surpasses the merged model accuracy for all cases at $p_d = 10\%$. Note that in this experiment, the cases with $p_d = 0\%$ belong to $\mathcal{AS}[100]$ and $\mathcal{AS}[110]$ for unknown and known model scenarios, respectively. For those cases, the adversary trains the clone model using its own dataset. Therefore, we observe that the clone model accuracy under the known model scenario (i.e., $\mathcal{AS}[110]$) is higher than that under the unknown model scenario (i.e., $\mathcal{AS}[100]$). This is because the model is randomly initialised in $\mathcal{AS}[100]$, whereas the initial model in $\mathcal{AS}[110]$ is the pre-trained model with higher accuracy on the target task. The merged model accuracy remains consistent for all $p_d$, as it only depends on the number of layers merged, given the same fine-tuned models.

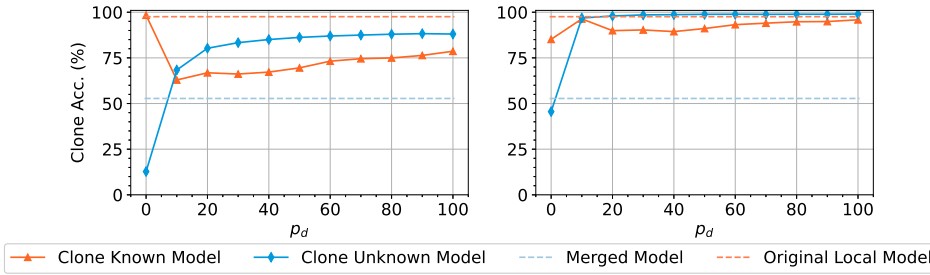

Figure 8: Clone accuracy for different proportions of data samples ($p_d$) in EuroSAT→DTD (left) and DTD→EuroSAT (right) scenarios.

## 5 RELATED WORK

Model merging (aka model fusion) is a technique that combines the parameters of models with different capabilities to build a single multi-task model (Yang et al., 2024a). A typical approach to construct a multi-task model using the model merging technique is to use a common pre-trained model as a backbone and merge the fine-tuned models for different downstream tasks (Matena & Raffel, 2022; Ilharco et al., 2023). To reduce the resource consumption in the conventional FMM, PMM (Stoica et al., 2024) was proposed to merge only a subset of the layers in a model. However, despite the fact that one of the main purposes of PMM is to reduce model-privacy risks, previous empirical studies on PMM mainly focused on resource reduction and performance optimisation, and the privacy protection perspective of PMM remains unexplored.

Model stealing is a group of privacy attacks targeted at NN models where the adversary aims to construct an alternative NN model that behaves similarly to the victim model. Papernot et al. (2017) proposed a model stealing attack based on the assumption that the adversary can access a subset of training data, but the model structure is unknown to the adversary. The adversary trains an alternative model with similar decision boundaries as the victim's model using a synthetic dataset (Papernot et al., 2017). The dataset is generated based on the accessible subset of the data using a technique named Jacobian-based Dataset Augmentation (JBDA) (Papernot et al., 2017). Alternatively, Orekondy et al. (2019) proposed a model stealing attack model where the adversary aims to steal the functionalities of the victim model. Their technique, named "Knockoff" (Orekondy et al., 2019), is based on a black-box assumption similar to (Papernot et al., 2017). However, the adversary in (Orekondy et al., 2019) cannot access any of the training data samples, and an alternative set of data is used to train the "Knockoff" model. Roberts et al. (2019) showed that it is also possible to perform model stealing attacks using only randomly generated data samples, given that the adversary has knowledge of the victim model's structure. However, none of the existing model stealing attacks is targeted at stealing the partial model's behaviour, given only a part of the victim model.

## 6 CONCLUSION

In this paper, we proposed and analysed a model stealing attack in PMM. The adversary can perform the attack with different prior knowledge, including the victim's shared model parameters, private model structure and training data samples. We performed attack simulations to compare our proposed attacks with existing model stealing attacks, with the same assumption about the adversary's prior knowledge. We showed that our attack is more successful than the baseline attacks in most of the scenarios we considered. We also explored our proposed attack with various numbers of private layers and data leakage and formalised a layer selection process in Appendix H. Results show that keeping fewer layers private can improve the merged model's performance at the cost of a higher attack success rate. Only a small fraction of data leakage can help the adversary achieve a better attack performance. Furthermore, we showed that the adversary can leverage a deep-shallow model structure to simulate the behaviour of an unknown model with similar or higher performance compared to the original model.

## THE USE OF LARGE LANGUAGE MODELS (LLMS)

We used large language models (ChatGPT, Gemini, *etc.*) as the general-purpose assistive tool during the preparation of this paper. Its contributions were limited to improving grammar, polishing wording, and suggesting alternative phrasings for clarity and conciseness. The research ideas, methodological design, experimental implementation, analysis, and final interpretations were entirely conceived and executed by the authors.

LLMs were not used for generating novel research content, fabricating facts, or conducting scientific reasoning. All technical descriptions, results, and conclusions presented in the paper are the sole responsibility of the authors.

## REPRODUCIBILITY STATEMENT

We have made every effort to ensure the reproducibility of our work. The details of the model architecture, training objectives, and hyperparameters are provided in Appendix B of the main paper. A complete description of the experimental setup, including datasets, preprocessing steps, and evaluation metrics, is included in Appendix D and Section 4.1.

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

# A    MODEL ACCURACIES FOR VIT-B/32, VIT-B/16 AND VIT-L/14 MODELS

Table 2: Model accuracies of the fine-tuned models, fully-merged model (FMM), and partially-merged model with 75% merged layers (PMM).

| | DTD | | | EuroSAT | | |
|---|---|---|---|---|---|---|
| Model | ViT-B/32 | ViT-B/16 | ViT-L/14 | ViT-B/32 | ViT-B/16 | ViT-L/14 |
| Parameters per Layer | 7087872 | 7087872 | 12596224 | 7087872 | 7087872 | 12596224 |
| Fine-tuned (DTD) | **97.55%** | **98.14%** | **98.24%** | 35.00% | 34.07% | 56.30% |
| Fine-tuned (EuroSAT) | 34.52% | 35.53% | 47.61% | **99.85%** | **99.89%** | **99.93%** |
| FMM | 61.44% | 64.04% | 77.87% | 76.41% | 79.67% | 95.89% |
| PMM | 52.77% | 54.84% | 72.50% | 67.67% | 79.11% | 95.04% |
| PMM/FMM | 85.89% | 85.63% | 93.10% | 88.56% | 99.30% | 99.11% |

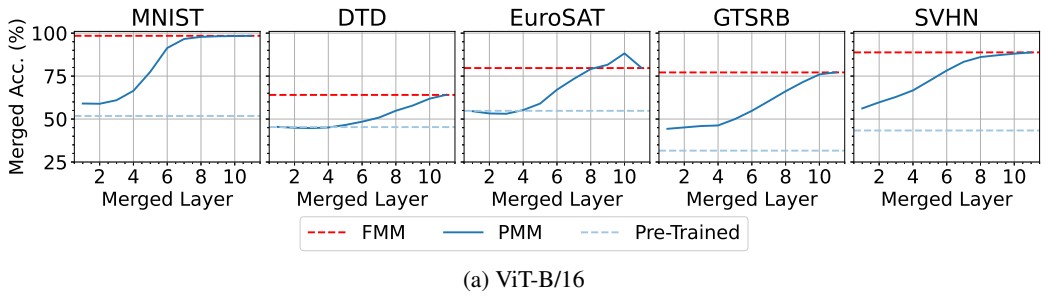

(a) ViT-B/16

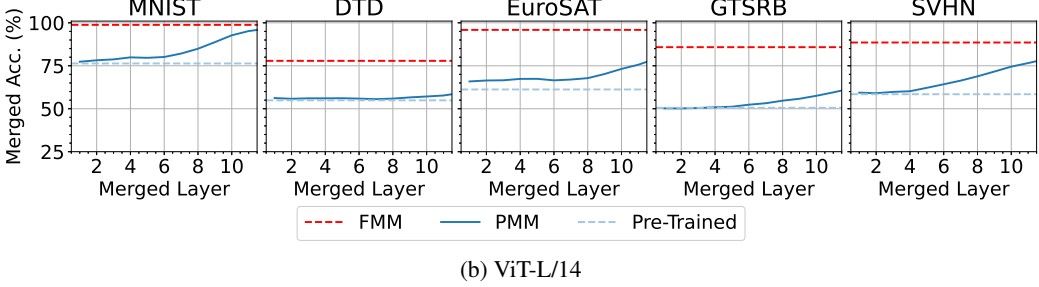

(b) ViT-L/14

Figure 9: Merged model accuracy for the partially merged model with different numbers of merged layers.

# B    DEFAULT EXPERIMENTAL SETTINGS

We use a workstation equipped with an Intel Xeon Gold 6248R CPU and two NVIDIA RTX A5000 GPUs. The memory size is 128 GB. The NN model training and merging are based on the PyTorch Python library. We set the number of training rounds to be 1500 after performing some trial runs to ensure that the model has converged by round 1500.

Unless otherwise stated in the experimental results, we use the following default hyperparameters:

- Target model structure: ViT-B/32
- The victim's shared model part is known: $\mathbb{I}_s = 1$
- Known data (i.e., $\mathbb{I}_d = 1$): 10% of the data samples in the adjusted dataset (i.e., $p_d = 10\%$);
- Sub-model weights for $\mathbb{I}_p = 0$: $\alpha = \beta = 0.5$;
- Deep sub-model (i.e., $M_1$) for $\mathbb{I}_p = 0$: multilayer LSTM;
- Learning rate for the clone model: 1e-5 for $\mathbb{I}_p = 0$ (deep-shallow model), 0.001 for $\mathbb{I}_p = 1$ (original model structure);

## C  RESULTS ON THE T5 MODEL

To assess the generalisation, we extend our evaluation for the *ModelPirate* attack beyond vision transformers and classification tasks using the T5-based encoder–decoder large language model (LLM) architecture (Raffel et al., 2020), which differs substantially in structure and complexity from the previously evaluated vision transformers and image classification tasks. Specifically, we conducted experiments on two NLP tasks – sentiment analysis on the IMDB dataset (Maas et al., 2011) and question answering on the QASC dataset (Khot et al., 2019). The fine-tuned models are publicly available at Hugging Face [6] [7].

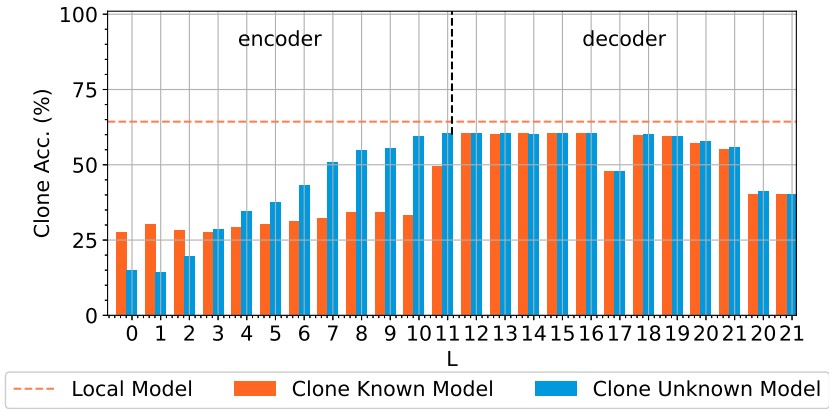

Figure 10: Clone accuracy for different layers in T5 model.

In Fig. 10, we present the results using IMDB sentiment analysis as the source task and QASC question answering as the target task. The results demonstrate that *ModelPirate* can effectively replicate the target model's behaviour and construct a clone model with similar model accuracy to the target model, especially for $l$ between 12 and 16 within the decoder module of the T5 model.

## D  DATASETS USED FOR THE EXPERIMENTS

The datasets used for the main experiments are listed below in Table 3.

Table 3: Datasets used for the main experiments.

| Dataset | Classification Task | Classes | Original Samples | Avg. | Adjusted Samples | Avg. |
|---|---|---|---|---|---|---|
| MNIST (Lecun et al., 1998) | Handwritten digits | 10 | 60000 | 6000 | 2000 | 200 |
| DTD (Cimpoi et al., 2014) | Textural image | 47 | 5640 | 120 | 5640 | 120 |
| EuroSAT (Helber et al., 2019) | Land use and cover | 10 | 27000 | 2700 | 2700 | 270 |
| GTSRB (Stallkamp et al., 2012) | Traffic light | 43 | 51840 | 1206 | 5184 | 121 |
| SVHN (Netzer et al., 2011) | House number digits | 10 | 99289 | 9929 | 3310 | 331 |

## E  EXPERIMENTAL RESULTS FOR MNIST AND SVHN DATASETS

### E.1  COMPARISON FOR MODEL STEALING ATTACKS AT DIFFERENT LAYERS

---

[6] https://huggingface.co/mrm8488/t5-base-finetuned-imdb-sentiment
[7] https://huggingface.co/mrm8488/t5-base-finetuned-qasc

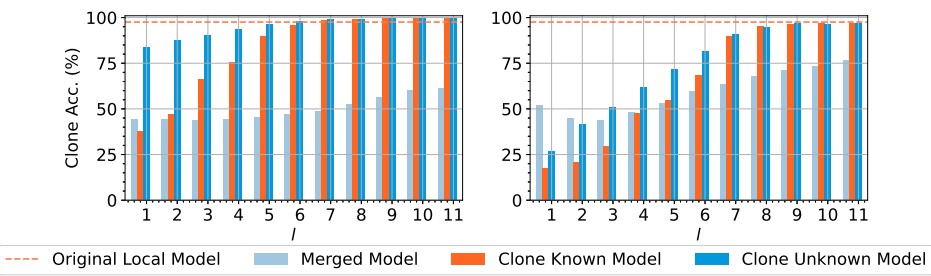

Figure 11: Clone accuracy for different numbers of merged layers $l$ in MNIST→SVHN (left) and SVHN→MNIST (right) scenarios.

### E.2 COMPARISON FOR MODEL STEALING ATTACKS WITH DIFFERENT PROPORTIONS OF KNOWN DATA SAMPLES

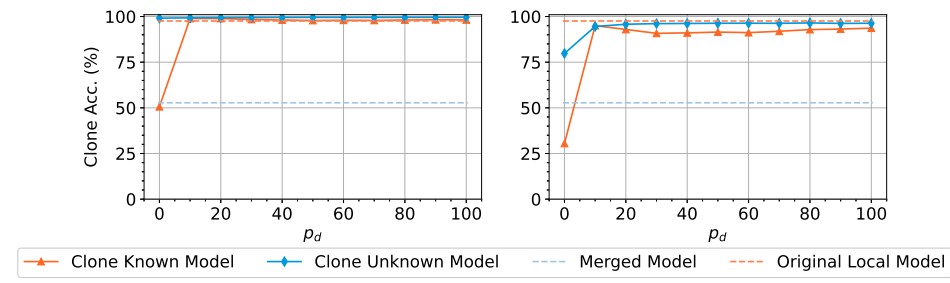

Figure 12: Clone accuracy for different proportions of data samples ($p_d$) in MNIST→SVHN (left) and SVHN→MNIST (right) scenarios.

## F EXPERIMENTAL RESULTS FOR $\mathbb{I}_p = 0$ WITH DIFFERENT $P_v^*$ SUB-MODEL STRUCTURES

In previous experiments, we only considered a pair of source and target tasks (i.e., DTD and EuroSAT). We repeat those experiments for an alternative pair of source and target tasks and show a similar trend to the results in the previous sections. Then, we further expand the experiments for $\mathbb{I}_p = 0$ for all source and target tasks with the rest of the parameters set as defaults. To analyse the impact on the clone model accuracy by the deep and shallow sub-models (i.e., $M_1$ and $M_2$ in Fig. 6), we remove $M_1$ or $M_2$ and repeat the simulation subsequently. From Table 4, we see that the clone model $P_v^*$ with all sub-models $M_1 + M_2 + M_3$ yields higher clone accuracy than $M_1 + M_3$, and similar clone accuracy as $M_2 + M_3$. Based on this observation, we conclude that the shallow model $M_2$ contributes more to the clone model's accuracy than the deep model $M_1$.

## G FINE-GRAINED LAYERS IN A VIT RESIDUAL BLOCK

In previous experiments, we only considered separating the model after a residual block in the ViT model. We conduct experiments to investigate how merging different components within a ViT residual block affects the effectiveness of *ModelPirate*. We repeat the experiments for separating after the attention block, feed-forward block, and the entire residual block at $l = 7$. As shown in Fig. 13, the attack remains effective across all configurations for two benchmark tasks, with nuanced variations in clone accuracy depending on which sub-component is shared.

Table 4: Clone Accuracy for different $P_v^*$ model structures and target tasks.

(a) $\mathcal{AS}[100]$

|  | MNIST | DTD | EuroSAT | GTSRB | SVHN |
|---|---|---|---|---|---|
| $M_1 + M_2 + M_3$ | 99.20% | 67.93% | 97.37% | 90.87% | 94.66% |
| $M_2 + M_3$ | 99.21% | 67.18% | 97.41% | 91.09% | 94.85% |
| $M_1 + M_3$ | 97.99% | 55.32% | 87.67% | 76.29% | 85.91% |

(b) $\mathcal{AS}[101]$

|  | MNIST | DTD | EuroSAT | GTSRB | SVHN |
|---|---|---|---|---|---|
| $M_1 + M_2 + M_3$ | 42.57% | 5.11% | 33.31% | 11.67% | 49.25% |
| $M_2 + M_3$ | 42.19% | 5.32% | 34.14% | 13.40% | 49.78% |
| $M_1 + M_3$ | 30.36% | 3.66% | 24.09% | 5.15% | 29.94% |

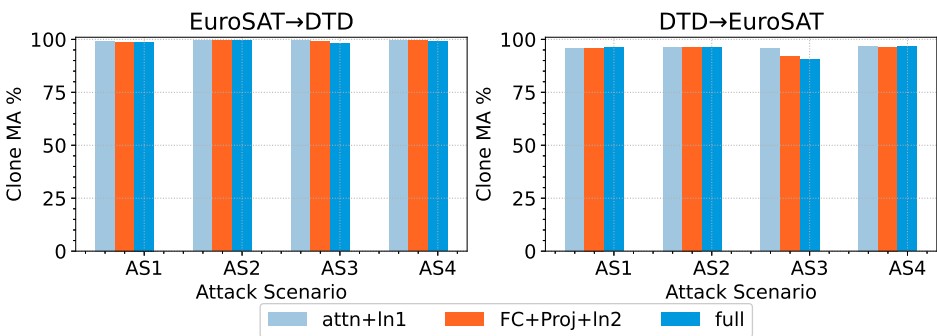

Figure 13: Clone accuracy for different fine-grained layers in ViT residual block eight.

## H   LAYER SELECTION GUIDELINE

We formalise the layer selection process as an optimisation problem that balances privacy leakage and performance gain. Specifically, a client can determine the optimal number of private layers by minimising a composite objective that incorporates (i) layer-wise information exposure, measured via auxiliary loss on neuron activations, and (ii) performance improvement, quantified by incremental gains across layers. The trade-off is controlled by a user-defined scaling factor. Let the optimal number of private layers be $l^*$. Then, we have

$$\underset{L^*}{argmin} \frac{1}{L} \sum_{l=1}^{L} \left( (1 + l * \epsilon) \times \phi_l - \lambda \Delta p_l \right) \tag{7}$$

Where:

- $\epsilon$ represents the incremental privacy leakage per layer. Theoretically, the cumulative privacy exposure increases non-linearly as the increase in the number of layers shared due to the additional information embedded in the combined layers compared to individual layers;
- $\phi_l$ is the information carried by all neurons in layer $l$, estimated via an auxiliary loss on the neuron activations. A higher auxiliary loss implies that the activations contain more informative (and potentially sensitive) content;
- $\Delta p_l$ is the performance gain by the layer. It can be computed as the difference in the model performance by re-training a partial model, up to layers $l$ and $l-1$; and
- $\lambda$ is the scaling factor determined by the clients to balance the privacy loss and performance loss measurements.

## I    COMPUTATIONAL AND COMMUNICATION OVERHEADS

We measure the change in computational and communication overheads with different numbers of layers merged. Let the time consumption for merging $l$ out of $L$ layers be $t_l$. The average increase in computational and communication costs when an additional layer is merged is calculated as:

$$100\% \times \frac{\sum_{l=2}^{L} \frac{t_l - t_{l-1}}{t_{l-1}}}{L - 1}. \tag{8}$$

We summarise the results in Table 5. The results indicate that each additional layer shared increases the computational and communication costs by approximately 3.83% to 7.23%, depending on the model structure. The results in Table 5, together with Fig. 2, demonstrate a trade-off between privacy preservation and model utility.

Table 5: Computational and communication costs when an additional layer is merged.

| Model | ViT-L/14 | ViT-B/16 | ViT-B/32 |
|---|---|---|---|
| Overhead | 3.83% | 7.23% | 6.41% |

