# OpenReview forum: "ModelPirate: Security Analysis of Partial Merging Against Model Stealing Attacks"
_ICLR.cc/2026/Conference — ICLR 2026 Conference Withdrawn Submission_

### Official Review · Reviewer_8tdM · 2025-10-24

**Soundness:** 2
**Presentation:** 2
**Contribution:** 2
**Rating:** 2
**Confidence:** 4

**Summary:**

This paper investigates model stealing attacks in the setting of partial model merging. They investigate how the efficacy of model stealing attack changes with varying number of shared layers, training data availability and knowledge of model structure. Their experiments show that model merging without adequate protection is highly vulnerable to model stealing attacks with clone accuracies exceeding the accuracy of the merged models.

**Strengths:**

1. The key ideas are presented well.
2. The experimental methodology seems well thought out and clearly ablates the effects of different components of the attack.
3. The deep-shallow design for the clone model is interesting.

**Weaknesses:**

1. In the introduction (L052), authors claim the partial model merging emerged in response to the problem of model stealing attack. I’m not convinced of this premise.
This claim is not supported by the ZipIt! paper that proposed partial model merging.
2. There are way too many notations, which make it hard to read the paper. A table which describes the notations might be useful.
3. I did not find much novelty in the design of the model stealing attack itself (besides the deep-shallow design of the clone model). The paper simply applies existing model stealing methods to the setting of partial model merging and provides results.
4. Experiments are limited to two datasets.

**Questions:**

1. Can you provide citations to support the claim that partial model merging is supposed to help with model stealing.
2. L261: How does the adversary have access to the classification head T_v? Isn’t this privately held by C_v?
3. How does the efficacy of the attack change if you don’t adopt a deep-shallow design?

---

### Official Review · Reviewer_rMzP · 2025-10-25

**Soundness:** 3
**Presentation:** 3
**Contribution:** 3
**Rating:** 6
**Confidence:** 2

**Summary:**

This paper investigates the privacy vulnerabilities of Partial Model Merging (PMM), an emerging model merging paradigm that shares only a subset of parameters among clients to balance utility and privacy. It proposes ModelPirate, a novel model stealing attack specifically designed for PMM, which attempts to reconstruct the unshared private model part under eight attack scenarios with varying degrees of adversarial prior knowledge (training data, model structure, and shared parameters). Extensive experiments on both vision (ViT-based) and language (T5) models show that PMM is vulnerable to model reconstruction even with limited exposure.

**Strengths:**

- **Novel problem setting**: As far as I know, it is the first systematic privacy analysis of PMM, a timely and underexplored topic given the increasing adoption of model merging methods.
- **Comprehensive attack framework**: The formulation of eight attack scenarios with different knowledge configurations provides a detailed and rigorous threat model.
- **Thorough experiments**: Evaluation across multiple datasets, model architectures (ViT-B/32, ViT-B/16, ViT-L/14, T5), and modalities (vision and NLP) demonstrates good generality.

**Weaknesses:**

- **Lack of defense-side experiments.** The paper convincingly demonstrates the vulnerability of PMM but does not evaluate any defense baselines. It would be valuable to test whether common countermeasures, such as differential privacy (DP-SGD), weight quantization, or parameter noise injection, can mitigate the ModelPirate attack while maintaining model utility.
- **Limited ablation on merging algorithms.** All experiments are conducted with layer-wise Task Arithmetic. To verify the generality of the attack, it is necessary to repeat key experiments using alternative model merging algorithms, such as Fisher-weighted averaging, Adamerging, or TIES-merging. This would reveal whether ModelPirate is robust across different merging schemes.
- **Shallow evaluation on LLMs.** The extension to T5 is appreciated, but the analysis for NLP models is limited to a single pair of tasks (IMDB $\rightarrow$ QASC) and lacks deeper investigation of linguistic behaviors or generalization differences.
- **Insufficient analysis of loss function choices.**: The attack optimization is fixed to the MAE. Conducting ablation studies with alternative objectives such as cross-entropy, KL divergence, or knowledge-distillation loss, would clarify whether ModelPirate’s performance critically depends on this specific loss function and how robust it is across optimization objectives.

**Questions:**

Refer to Weaknesses.

---

### Official Review · Reviewer_vKhS · 2025-10-31

**Soundness:** 2
**Presentation:** 2
**Contribution:** 2
**Rating:** 2
**Confidence:** 4

**Summary:**

This paper investigates the privacy risks of Partial Model Merging (PMM), a method where clients merge only part of their fine-tuned neural networks to form a shared model while keeping some layers private. The authors introduce ModelPirate, a model-stealing attack that reconstructs the private portion of a victim model under eight adversarial scenarios with varying levels of prior knowledge, including shared model parts, private structure, and partial training data. Experiments on vision and language models, such as ViT, CLIP, and T5, across several datasets demonstrate that an attacker can recover a significant portion of the victim’s performance even with limited information.

**Strengths:**

The paper is clearly written and well structured. It defines a systematic threat model that enumerates adversarial capabilities in a comprehensive way. The experimental section is extensive, spanning multiple architectures and modalities, and the results are reproducible and well documented.

**Weaknesses:**

1. The conceptual scope of PMM needs clarification. PMM is not a well-established research paradigm, and the only cited related work, ZipIt! (Stoica et al., 2024), does not actually introduce PMM as a distinct framework or address privacy concerns. This makes the motivation for focusing on PMM somewhat speculative. Without broader evidence that PMM is being used or seriously considered in real systems, the practical relevance of the proposed analysis is limited.

2. The setup presented in Figure 3 is essentially equivalent to one-shot federated learning or split learning. In these established paradigms, each client fine-tunes a model locally, shares a subset of layers for aggregation, and then combines the result with its private part. The paper does not clearly explain how PMM differs conceptually or operationally from these methods. As a result, the framing of PMM as a distinct problem domain appears overstated, and the novelty of the contribution becomes less convincing.

3. The technical contribution of the ModelPirate attack is limited. The attack primarily applies standard model-stealing and knowledge-distillation principles under different assumptions about the adversary’s prior knowledge. The methodological novelty lies mainly in systematically enumerating these scenarios rather than introducing a new algorithmic mechanism. Consequently, the attack design feels more heuristic than innovative.

4. The paper works on a strong assumption that an adversarial client can query the victim’s full model to obtain labels for clone training. This assumption is reasonable in machine-learning-as-a-service settings, where an external user interacts with a deployed model via an API, but it contradicts the principles of PMM, where private parts of the model are never shared or exposed. Allowing cross-client querying effectively converts the setup into a model-extraction attack against a public service rather than a PMM-specific vulnerability. Without this assumption, the proposed attack would not function as described, raising doubts about the realism of the threat model.

5. The evaluation, while extensive, remains largely heuristic. It measures privacy leakage only through clone accuracy and does not provide a theoretical or information-theoretic interpretation of leakage. There is no analysis of how query budgets, output formats, or noise in victim responses would affect attack success. The experiments use small-scale datasets and fixed hyperparameters, with little justification for these design choices. While the results demonstrate that the attack can reproduce model behavior empirically, they do not provide insight into the mechanisms driving the leakage or its robustness under realistic constraints.

6. The comparisons rely on classic model-stealing baselines such as Knockoff, JBDA, and Random; omitting more recent or adaptive extraction techniques limits the strength of the evaluation. The paper also does not propose or evaluate any defenses, leaving the analysis one-sided and incomplete.

**Questions:**

1. How is the adversary realistically able to query another client’s full model within PMM?
2. Would the ModelPirate attack still apply if such querying were disallowed, and how would labels be obtained in that case?
3. How does PMM fundamentally differ from one-shot federated or split learning, which appear to follow the same workflow?
4. What aspects of ModelPirate are technically novel compared with existing model-stealing and distillation attacks?
5. Have you considered any concrete defense strategies, such as noisy merging, differential privacy, or secure aggregation?

---

### Official Review · Reviewer_Xx9S · 2025-11-01

**Soundness:** 2
**Presentation:** 1
**Contribution:** 2
**Rating:** 4
**Confidence:** 3

**Summary:**

This paper proposes ModelPirate, a framework for analyzing model stealing attacks aimed at reconstructing the private component of a model in a partial model merging scenario. In particular, the authors investigate how different types of information leakage including training data, client architecture, and the shared model component, affect the accuracy of model stealing. Through empirical experiments, the authors demonstrate that adversaries can still recover a substantial portion of the private model component even when they have access to only a small subset of the training data, the client architecture, and the shared part model.

**Strengths:**

- This paper provides insight into the trade-offs between different types of information leakage and their corresponding privacy risks in partial model merging.

- ModelPirate demonstrates high attack accuracy across multiple attack scenarios.

**Weaknesses:**

- The writing needs improvement and further proofreading. Some terms, such as local accuracy, are defined but not used throughout the paper.

- Regarding the evaluation metric, it is unclear why Merged Accuracy is considered the lower bound accuracy of the cloned model. If it truly represents a lower bound, then Table 1 should not contain results below this value; however, several entries do fall below it. Furthermore, the defining Merged Accuracy as the model accuracy of $S_m + P_a$ requires clarification. Since  $P_a$ corresponds to a different task, its classification head may output a different set of classes from those in  $T_v$, resulting in a different number of output classes. Even when the number of classes coincides, changing the label order in $T_v$  could still alter the model accuracy of  $S_m + P_a$

- The setup for training the client model using both the attack dataset  $D_a$ and the victim dataset $D_v$ is unclear. Because $D_a$ and
$D_v$ are distinct datasets with different label spaces, how are the labels in $D_a$ redefined or aligned for joint training of the new client model?

- I am curious about the role of the adversary in this model stealing setup.  In this setup, the adversary also acts as an ordinary participant, contributing its parameters to the merged model. What is the role or impact of this action? What would happen if the adversary did not participate in the merging process but still had access to certain information, such as part of the training data, the client model architecture, or the shared component of the client model?


- In lines 361–364, the authors argue that it is more important for a client to protect its training data than its private model structure. However, the results of AS[X01] and AS[X10] do not support this claim. For instance, in AS[001] and AS[010], the results for ViT-L/14 are contradictory. Notably, in AS[010], where only the private model structure is known, the attack surprisingly achieves 93.81%, the highest performance among all attacks on the ViT-L/14 model.

- The results in Table 1 also exhibit an abnormal trend: AS[110] often outperforms AS[111] by a large margin, implying that access to private training data may decrease attack performance. Moreover, AS[111] is not the best-performing setup, even though the adversary in this case knows the client model architecture, shared components, and training data. This raises concerns about whether the client model’s training process might limit its learning capability, thereby leading to misleading comparisons.

**Questions:**

- Could the authors clarify the definition of merged accuracy and explain how the adversary’s dataset $D_a$ is used to train the cloned client model?

- Could the authors elaborate on their argument that it is more important for a client to protect its training data than its private model structure?

- Could the authors provide more details about the client model’s training process and explain why the AS[111] setup does not yield the best performance?

- Could the authors to clarify the specific role of the adversary in the proposed model stealing setup when they also involve in merging process?

---

### Note · Authors · 2026-01-23

I have read and agree with the venue's withdrawal policy on behalf of myself and my co-authors.